# Neutron Interferometer Experiments Studying Fundamental Features of Quantum Mechanics

Armin Danner [1,*] , Hartmut Lemmel [1,2] , Richard Wagner [2] , Stephan Sponar [1] and Yuji Hasegawa [1,3,*]

1   Atominstitut, TU Wien, Stadionallee 2, 1020 Vienna, Austria; stephan.sponar@tuwien.ac.at (S.S.)
2   Institut Laue Langevin, 71 Avenue des Martyrs, 38000 Grenoble, France
3   Department of Applied Physics, Hokkaido University, Kita-ku, Sapporo 060-8628, Japan
*   Correspondence: armin.danner@tuwien.ac.at (A.D.); yuji.hasegawa@tuwien.ac.at (Y.H.)

**Abstract:** Quantum theory provides us with the best account of microscopic components of matter as well as of radiation. It was introduced in the twentieth century and has experienced a wide range of success. Although the theory's probabilistic predictions of final experimental outcomes is found to be correct with high precision, there is no general consensus regarding what is actually going on with a quantum system "en route", or rather the perceivable intermediate behavior of a quantum system, e.g., the particle's behavior in the double-slit experiment. Neutron interferometry using single silicon perfect crystals is established as a versatile tool to test fundamental phenomena in quantum mechanics, where an incident neutron beam is coherently split in two or three beam paths with macroscopic separation of several centimeters. Here, we present quantum optical experiments with these matter-wave interferometers, studying the effect of the quantum Cheshire Cat in some variants, the neutron's presence in the paths of the interferometer as well as the direct test of a commutation relation. To reduce disturbances induced by the measurement, the interaction strength is lessened and so-called weak interactions are exploited by employing pre- and post-selection procedures. All results of the experiments confirm the predictions of quantum theory; the observed behaviors of the neutron between the pre- and post-selection in space and time emphasize striking and counter-intuitive aspects of quantum theory.

**Keywords:** neutron interferometry; double slit; which-way; weak value; quantum Cheshire Cat; delayed-choice; uncertainty relations

## 1. Introduction

Quantum theory describes the fundamental and basic behaviors of systems at atomic and smaller scales [1–3]. The theory was developed intensively in the twentieth century and became the basis for modern technologies such as quantum electronics, solid-state engineering as well as the advent of laser and nuclear engineering. In contrast with these tremendous successes of great significance and value, the comprehensive view of the world provided by quantum theory is to some extent puzzling [4]. For instance, the classical theories can provide deterministic predictions, which we become used to in our everyday lives; the principles, which guide reasoning within a given circumstance à la quantum theory, are governed by a probability law. In order to accept what is claimed by quantum theory, one has to abandon classically employed ideas such as realism, causality and locality [5].

As stated by Feynman [6], the double-slit experiment serves as "a phenomenon which is impossible [...] to explain in any classical way, and which has in it the heart of quantum mechanics. In reality, it contains the only mystery [of quantum mechanics]". Currently, the double-slit experiment is demonstrated by using electrons [7–9], photons [10], ions [11,12], atoms [13–15], large molecules [16] and positrons [17]; the wave-particle duality in these experiments is viewed as quanta behaving both as waves and particles.

The neutron interferometer, made out of a silicon perfect-crystal of monolithic structure, was invented in 1974 [18]; it is an apparatus where particles, i.e., neutrons, exhibit wave properties in certain circumstances. The length of the interferometer is about 10 cm. The thermal neutrons used for all presented neutron optical experiments are characterized by energies of about 20 meV, velocities of about 2 km/s and wavelengths of about 2 Å. The maximum neutron flux exiting the interferometer is of the order of 100/s. Due to the given parameters of flux, velocity and diameter of the neutron interferometer, the vast majority of neutrons pass through the neutron interferometer alone, while in many cases the next detected neutron is still bound in the fuel of the reactor. Therefore, these experiments are in the domain of self-interference. The reactor is an incoherent neutron source, since the detected neutrons emerge from fissions that cannot be correlated to each other. The noteworthy character of the single-crystal neutron interferometer is the macroscopic dimension of both the beam separation and the area enclosed by the interfering beams, which are given by several centimeters and square centimeters, respectively. In addition, owing to this macroscopic separation of the beams, a variety of spin-manipulation equipment of macroscopic dimensions can be inserted in each beam path, allowing us to manipulate the neutron's degrees of freedom, i.e., spin as well as energy with high accuracy.

Neutron interferometry is established as one of the most fruitful approaches to study fundamental phenomena in quantum mechanics [19]. Just to mention a few, demonstrations of the $4\pi$-symmetry of a spinor wavefunction [20] and of a gravitationally induced phase shift [21] were achieved; by implementing the manipulation of spin in the interferometer, spinor superposition of a 1/2-spinor is clearly demonstrated [22]. More recently, entanglement between different degrees of freedom in a single particle, i.e., a neutron, is exploited to demonstrate peculiarities due to quantum contextuality; the violation of a Bell-like inequality is confirmed by the use of entanglement between two degrees of freedom [23], followed by further performances with multi-partite entangled states implemented in a single neutron [24]. Experiments investigating striking and counter-intuitive quantum mechanical phenomena on the fundamental level have been carried out [25,26].

While quantum theory gives the (probabilistic) prediction of the final result of a measurement performed on a quantum system, it is not clear whether one can say anything or what one can say on the value of a physical parameter of the system between the pre-selected initial state and the post-selected final state. A possible answer to this question is provided by the so-called weak value, which is proposed by Aharonov, Albert and Vaidman as a "new kind of value for a quantum variable" [27,28]. The weak value can be obtained operationally through the weak measurement, i.e., the weak interaction applied on the initial state and followed by the post-selection; the weak interaction is tuned so that the probed system evolves with minimal disturbance from the initial toward the final state. As a result of the weakness of the interaction, the shift of the meter (in the system in a measurement) is so small that the results of lots of trials should be accumulated to explicitly distinguish the results of the measurement. The obtained weak value in this procedure may have different features than the expectation values of the relevant observable and the states; weak values may have imaginary contributions and lie outside the eigenvalue range of the associated operator. Nevertheless, due to the weakness of the interaction, disturbances by the measurement, which are inevitably induced by a majority of quantum measurement, can be reduced considerably; weak values can be obtained to estimate the value of a physical parameter in an intermediate circumstance between the initial and the final state. The use of weak values developed from a theoretical peculiarity to their powerful experimental implementation [29–32].

In this review, recent accomplishments with the neutron interferometer are presented, which take advantage of the weak interactions and weak measurements to evaluate and study intermediate circumstances, i.e., quantum dynamics, of neutrons in the interferometer. Among many, three main issues are selected: first, the phenomenon of the quantum Cheshire Cat is presented in Section 2, including the initial experimental observation [33], the delayed-choice implementation [34], the extension to a three-path interferometer [35]

and the exchange of properties in a photonic system. Second, a new approach to the path presence measurement is presented [36] in Section 3. Third, a direct test of the commutation relation [37] is presented in Section 4. Section 5 discusses the meaning of weak values before the conclusion of Section 6.

## 2. Quantum Cheshire Cats

*"Well! I've often seen a cat without a grin,"* thought Alice *"but a grin without a cat! It's the most curious thing I ever saw in my life!"* Lewis Carroll's Alice has many a strange encounter in Wonderland [38]; however, it still boggles her mind when she meets the Cheshire Cat, whose body can disappear while its grin remains. A similar effect can be produced in quantum mechanical experiments [39]: the particle and spin property of neutrons can "appear to be separated in different paths" of a Mach–Zehnder interferometer. The spin is identified with the grin of the cat, while the particle is identified with the cat's body such that the perception given in Figure 1 is possible. This effect was demonstrated by Denkmayr et al. [33]. Here, we describe four experimental realizations of the quantum Cheshire Cat: the initial experiment and two extensions of it, one regarding a delayed choice of the separation, another separating a third property, and the exchange of grins in a photonic system. The description will use a particle picture to underline the counter-intuitive aspect of the observed phenomena. This interpretation uses weak values to "infer the locations of properties". In an alternative interpretation, weak values only quantify the reactions of the intensity to weak interactions. Then, the reactions occur because of the coherent superposition of multiple sub-states, which are recombined at the post-selection. This interpretation accentuates the wave property in the interferometer such that all involved properties are present in all paths. The weak values then characterize a specific post-selection as discussed in Section 5.

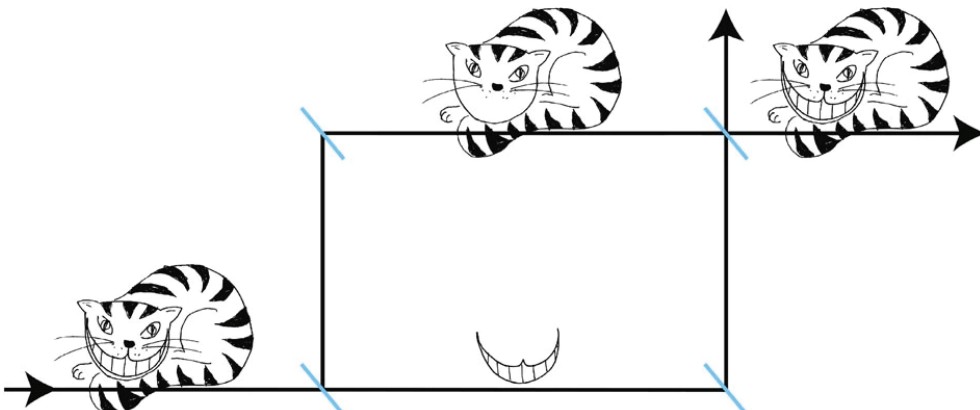

**Figure 1.** Schematic of a quantum Cheshire Cat where the properties of the neutrons are identified with parts of the cat. The cat's body represents the particle and its grin represents the spin. Manipulations, each "identifying the location of a different property", only yield effects in a different path of the interferometer such that one may perceive "the properties to be separated" inside the interferometer.

### 2.1. Initial Quantum Cheshire Cat

The experiment was carried out at the S18 interferometer beam line at the high-flux research reactor of the Institut Laue-Langevin (ILL) in Grenoble, France. The applied neutron interferometric setup is depicted in Figure 2. The monochromatic neutrons are polarized by a magnetic birefringent prism, which deflects the spin-down (blue arrow in Figure 2) neutrons out of the Bragg acceptance angle of the interferometer crystal (typically a few seconds of arc). The direct-current (DC) spin turner DC1 in front of the interferometer rotates the neutron spin by $\pi/2$ into the $+x$ direction. Then, the incident beam is coherently split into two beams at the first plate of the interferometer. Two spin rotators (SRs) prepare the initial state $|i\rangle = 1/\sqrt{2}\,(|I, \rightarrow\rangle + |II, \leftarrow\rangle)$, where $\leftrightarrow$ denotes the eigenstates of the Pauli

spin matrix $\hat{\sigma}_x$. The absorber can be inserted in path I or II as a weak probe of the neutrons' presence in the respective path. The spin of the interfering beam in the forward direction is analyzed by the use of the coil DC2 together with a magnetic supermirror (analyzer), post-selecting on the final state $|f\rangle = 1/\sqrt{2}\,|\leftarrow\rangle\,(|I\rangle + |II\rangle)$.

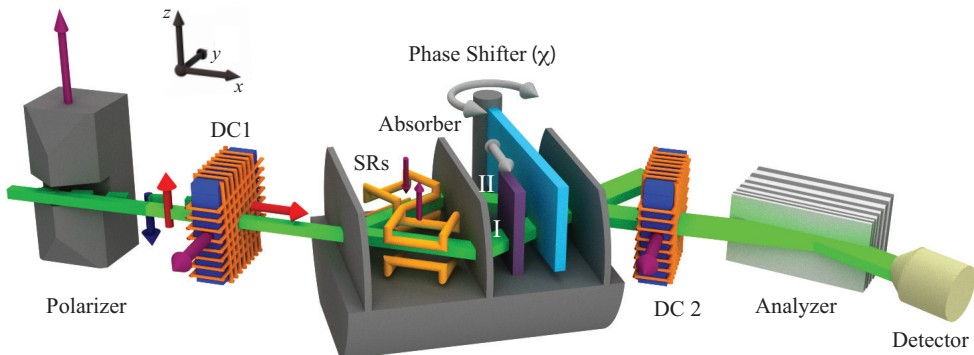

**Figure 2.** Illustration of the experimental setup of the initial quantum Cheshire Cat experiment. Purple arrows give the direction of local magnetic fields, gray arrows indicate the spatial motion—translation and rotation—of the absorber and phase shifter, respectively. Red and blue arrows are the initial up and down spin polarization vectors. The fraction that is initially up polarized is rotated by the field in the coil DC1 into the $x$ direction.

To determine the neutrons' population in the interferometer's paths, the weak values of the projection operator onto path $j$, denoted as $\langle \hat{\Pi}_j \rangle_{\mathrm{w}} = \langle f | \hat{\Pi}_j | i \rangle / \langle f | i \rangle$, are measured. First, a reference measurement is performed, where the orthogonal spin states of path I and II result in a constant intensity when rotating the phase shifter. Next, the weak absorber with transmissivity $T = 0.8$ is inserted into path I and the phase shifter scan is repeated, resulting in the same constant intensity. However, if the very same absorber is put in path II, the intensity decreases, suggesting that the neutrons' population in the interferometer is obviously higher in path II than in path I. The observed intensities for measurements with an absorber in path I and II as well as the reference measurement are depicted in Figure 3.

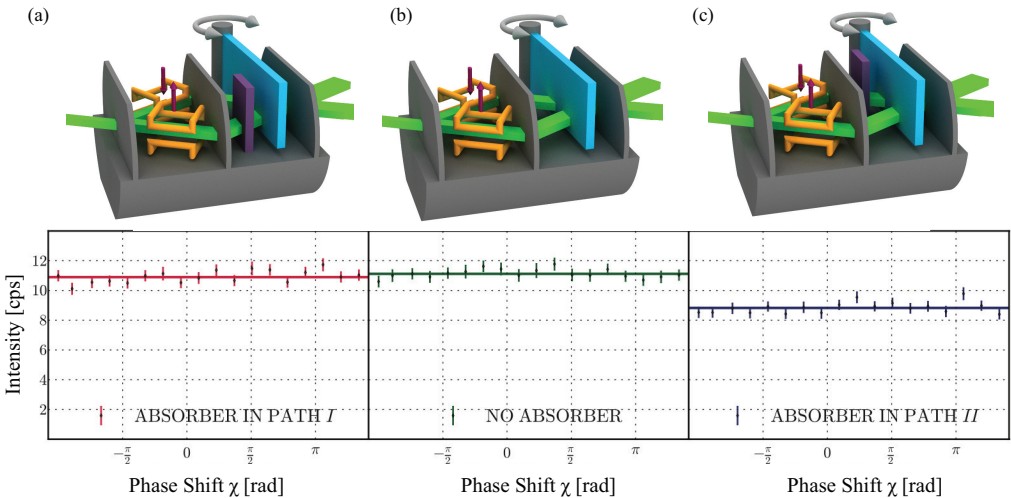

**Figure 3.** Setup and results of the neutrons' population in the initial quantum Cheshire Cat experiment. Purple arrows are the direction of local magnetic fields, gray arrows indicate the physical rotation of the phase shifter. An absorber is inserted (**a**) in path I or (**c**) in path II, while (**b**) the reference measurement is done without the absorber.

The weak measurements of the neutrons' spin component in each path are achieved by applying an additional weak magnetic field in path I or path II, causing small spin

rotation angles. This procedure allows us "to probe the presence of the neutrons' spin" in the respective path. The condition of a weak measurement is fulfilled by tuning the magnetic field to be sufficiently small; in this experiment, spin rotations of 20° were used. Such an additional magnetic field in path I leads to the emergence of interference fringes with a contrast of ≈0.3, indicating "the presence of the neutron's spin". On the other hand, the same field applied in path II causes no significant change in the intensity modulation. The measured intensities are seen in Figure 4.

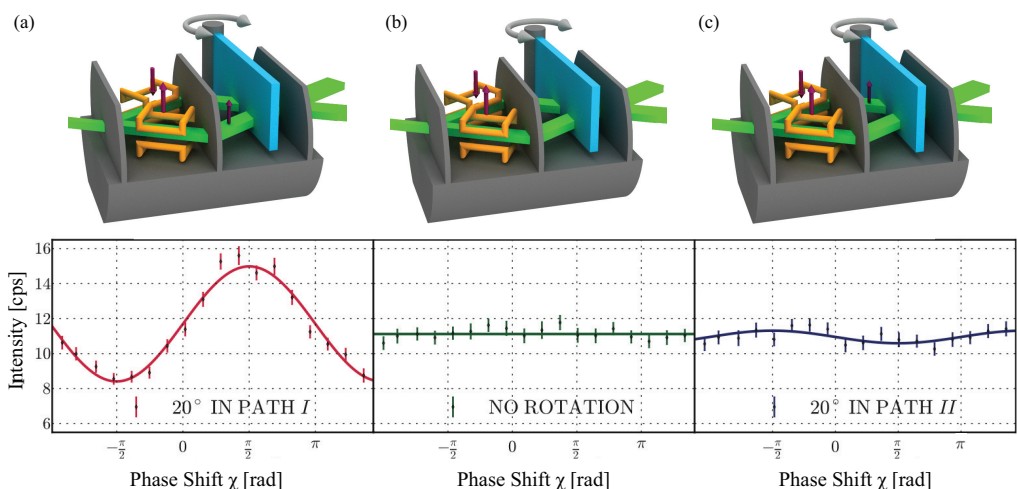

**Figure 4.** Setup and results of the neutrons' spin component in the initial quantum Cheshire Cat experiment. Purple arrows are the direction of local magnetic fields, gray arrows indicate the physical rotation of the phase shifter. A weak magnetic field is applied (**a**) in path I or (**c**) in path II, while (**b**) the reference measurement is done without the magnetic field.

From the obtained results, one can therefore conclude that the neutrons go through path II, while "the spin travels along path I", demonstrating the quantum Cheshire Cat effect.

### 2.2. Delayed-Choice Quantum Cheshire Cat

The essential feature of the quantum Cheshire Cat in the neutron interferometer is that "the cat itself and its grin are separated from each other and located in different paths". In such a setup, the following questions arise: what happens if the choice of the post-selection is postponed until the quantum particle already entered the interferometer? Will "the particle and its property still be separated spatially"? Or to pose an even more provoking question: can "the location of the separated particle and its property" be influenced by the choice of the post-selection, that has been decided upon afterwards? It allows us to study quantum causality, where the causal order of events may be undefined [40,41]. This follows a gedanken experiment already put forward by Wheeler [4,42]. In his proposed setup, the choice, whether to observe the wave property (interference) or the particle property (path), is delayed until photons have already traveled beyond the first beam splitter of an interferometer. The realization of several such experiments has been reported. An overview is given in [43]. A schematic view of the experimental setup is shown in Figure 5. The coil labeled DC2 is switched quickly and at random between the two possible states of post-selection, either $|f_+\rangle = 1/\sqrt{2}\,|\leftarrow\rangle(|I\rangle + |II\rangle)$ or $|f_-\rangle = 1/\sqrt{2}\,|\rightarrow\rangle(|I\rangle + |II\rangle)$. The switching is done at a rate so high that neutrons in the split beams of the interferometer do not know which post-selection will be applied, i.e., where "the particles and its properties should be located". The choice is so to speak delayed until the neutrons already coherently split into the two paths and started propagating through the interferometer.

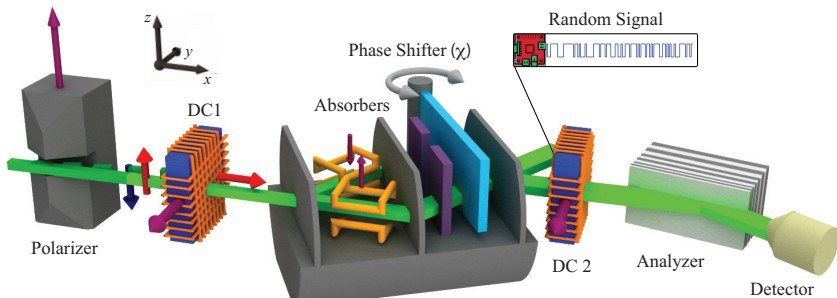

**Figure 5.** Experimental setup of the delayed-choice quantum Cheshire Cat experiment. Purple arrows give the direction of local magnetic fields, gray arrows indicate the spatial rotation of the phase shifter. Red and blue arrows are the initial up and down spin polarization vectors. The fraction that is initially up polarized is rotated by the field in the coil DC1 into the $x$ direction. The rotation angle of the spin turner DC2 switches randomly between the two values $\pm\pi/2$, realizing the post-selections of $|f_\pm\rangle$.

The final results of the experiment are depicted in Figure 6. It is seen that for the post-selection of $|f_+\rangle$, the interferogram is unchanged with the absorber in path I and only an absorber put in path II has an influence on the intensity such that the neutron is located in path II. Nonetheless, for the post-selection of $|f_-\rangle$, the intensity drop is swapped from path II to path I. By changing the post-selection to $|f_-\rangle$, the obtained results confirm that the position of the neutrons are now found in path I, while no effect is seen when putting the absorber in path II. It is concluded from the above results that the position of the neutrons' location depends solely on the choice of the post-selection, which is either $|f_+\rangle$ or $|f_-\rangle$. For the spin, it is the other way around: when "locating the spin" with weak spin rotations in path I, a significant increase in contrast is observed for the post-selection of $|f_+\rangle$. For the post-selection of $|f_-\rangle$, "the spin is found in path II". In the experiment it is hence confirmed that, although the cat (neutron) is not aware of these selections as it enters the interferometer, "the locations of the cat and the grin are interchanged" according to the delayed and random choice of the post-selection.

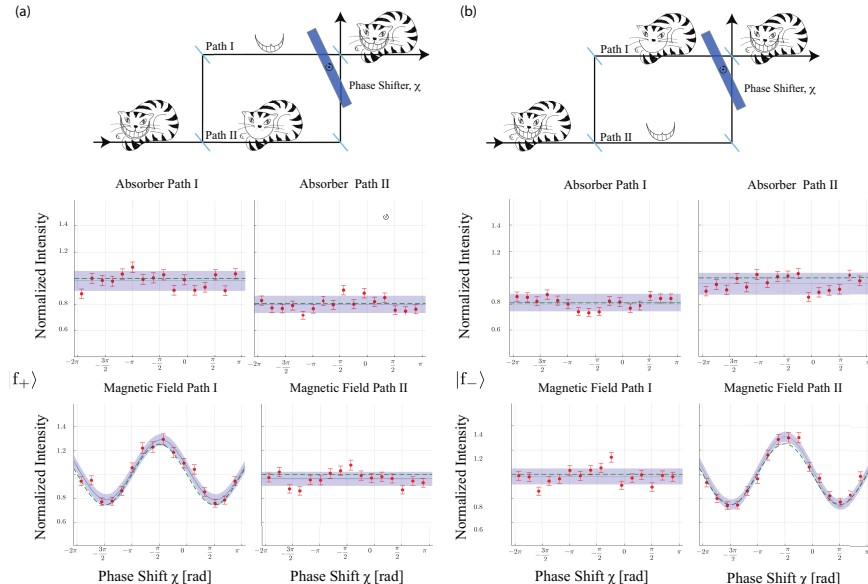

**Figure 6.** Graphical depiction of the emergence of the quantum Cheshire Cat with the delayed-choice implementation of the post-selected states (**a**) $|f_+\rangle$ and (**b**) $|f_-\rangle$. For both (**a**,**b**), the top schematic gives a possible impression of the below measurement results of intensities. The upper intensities are the result of weak absorber measurements and the bottom intensities the results of weak spin rotation measurements, while tuning the relative phase $\chi$ with the phase shifter.

### 2.3. Three-Path Quantum Cheshire Cat

In the initial description of the quantum Cheshire Cat effect [39], the authors asked whether two properties could simultaneously be "separated" from a particle. Pan [44] picked this up and developed the general case with arbitrarily many properties and paths. The three-path quantum Cheshire Cat with the three properties of spin, particle and energy realized with neutrons [35] follows this proposal. The properties "appear to be separated into the three paths" of an interferometer such that each property is located in a different sub-beam. Three parts of the neutron are "apparently separated", which are represented by three parts of a cat, such that the perception depicted in Figure 7 is possible. In Figure 7, the schematic of Figure 1 of the Cheshire Cat is extended by representing the additional manipulated energy system of the neutrons with the stripes of the cat. By using the energy as a third degree of freedom, the geometrical relations in Hilbert space between the state vectors of the sub-beams are investigated when weak interactions are applied.

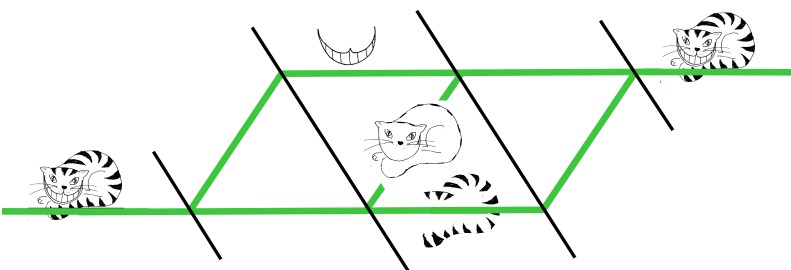

**Figure 7.** Schematic of a three-path quantum Cheshire Cat where the properties of the neutrons are represented with parts of the cat. The cat's body represents the particle, its grin represents the spin and its stripes the energy. Three different manipulations, each "identifying the location of a different property" of the neutron, only yield effects in a different path of the interferometer such that one may "perceive the three properties to be separated" inside the interferometer.

The paths of the interferometer are indexed as $j \in \{\text{I}, \text{II}, \text{III}\}$, the two different energy levels are referred to as $\text{E}_0$ and $\text{E}'$ and the $z$ spin eigenstates as $\uparrow$ and $\downarrow$. The pre-selected state $|\text{i}_{3\text{path}}\rangle$ is written as

$$|\text{i}_{3\text{path}}\rangle = \frac{1}{\sqrt{3}} \Big( |\text{I}, \downarrow, \text{E}_0\rangle + |\text{II}, \uparrow, \text{E}_0\rangle + |\text{III}, \downarrow, \text{E}'\rangle \Big) \tag{1}$$

where all states of different Hilbert spaces corresponding to a path are written together in a single ket. All sub-states of the single paths are mutually orthogonal in the spin or energy system. At the pre-selection, a value of spin and energy can be attributed to each sub-beam, although later on the properties may be "perceived to be separated". The post-selection is represented by the projector to the state $|\text{f}_{3\text{path}}\rangle$ given by

$$|\text{f}_{3\text{path}}\rangle = \frac{1}{\sqrt{3}} |\uparrow\rangle \left( e^{\text{i}(\chi_2 - \chi_1)} |\text{I}\rangle + e^{\text{i}(\chi_1 + \chi_2)} |\text{II}\rangle + e^{\text{i}(\chi_1 - \chi_2)} |\text{III}\rangle \right) \tag{2}$$

which does not contain any terms of the energy, meaning no energy selection is employed in the post-selection. Therefore, all neutrons of up spins and specific phase relations between the paths with arbitrary energy are selected to propagate towards the detector. The phases $\chi_1$ and $\chi_2$ are determined by the orientation of the phase shifters depicted in Figure 8. The overlap $\langle \text{f}_{3\text{path}} | \text{i}_{3\text{path}} \rangle$ has its sole contribution through the reference beam of path II.

Three different weak interactions are applied. Each weak interaction is supposed to affect a different property: an Indium foil with absorption coefficient $\mathcal{A}$ can absorb the particle; a direct-current (DC) spin rotation with rotation angle $\alpha$ changes the spin orientation; and a radio-frequency (RF) spin rotation with the same rotation angle $\alpha$ changes the energy of the neutron through its time-dependent magnetic field. The interactions are weak, meaning $\mathcal{A}, \alpha \ll 1$.

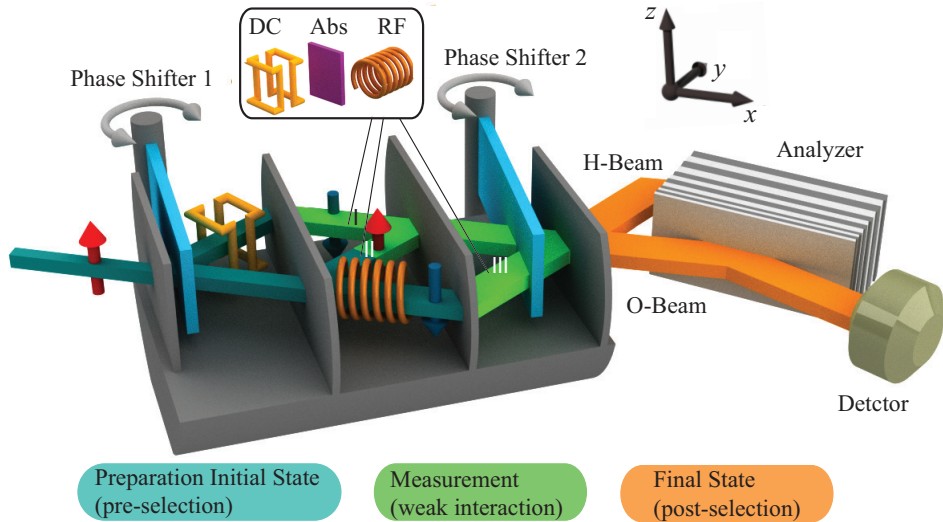

**Figure 8.** Setup of the experiment for the three-path quantum Cheshire Cat. Gray arrows indicate the spatial rotation of the two phase shifters. Red and blue arrows indicate the local polarization vectors before and after the preparation stage. The local polarization vectors are symbolized by red arrows for the up spin orientation and blue arrows for the down spin orientation. The setup contains a pre- and post-selection. In between, weak interactions can be applied while the reactions in the O-beam are monitored.

The experiment was carried out at the neutron interferometry instrument S18 at the Institut Laue-Langevin (ILL) in Grenoble, France. The setup downstream of the monochromator and polarizer is depicted in Figure 8. Since all sub-states of the pre-selection are mutually orthogonal either in the spin or energy system, no interference is expected to appear at the detector. Contrasts $C \approx 0$ are observed in the respective preparational interferograms, cf. the original paper [35]. When applying the weak interactions and recording the weak interaction interferograms, conspicuous differences to the preparational interferograms are observed in a different path for each interaction, as can be seen in Figure 9: an absorber only reduces the intensity in path II, a weak spin rotation produces interference fringes only when applied in path I, and a weak energy manipulation produces interference fringes only when applied in path III. These three statements are quantified by the respective path weak values $\langle \hat{\Pi}_j \rangle_\text{w}$, the spin weak values $\langle \hat{\sigma}_x^\text{DC} \hat{\Pi}_j \rangle_\text{w}$ and the energy weak values $\langle \hat{\sigma}_x^\text{RF} \hat{\Pi}_j \rangle_\text{w}$, where the latter two depend on the phase shifter orientation [35]. The absolute value of the weak values corresponding to the interferograms with yellow background are approximately 1, while the others are close to zero.

When "identifying the locations of the properties" through the conspicuous reactions given in Figure 9, a three-path quantum Cheshire Cat is generated. The "spin is located in path I", the particle in path II and "the energy in path III". It is undisputed that the path weak values $\langle \hat{\Pi}_j \rangle_\text{w}$ quantify through which path the pre- and post-selected neutrons went. This is connected with the behavior of the intensity $I$, which is exclusively linear to the absorption coefficient $\mathcal{A}$ of an absorber put in path $j$ such that

$$I_j^\text{Abs} = \left| \langle \text{f}_\text{3path} | \text{i}_\text{3path} \rangle \right|^2 \left[ 1 - \mathcal{A} \langle \hat{\Pi}_j \rangle_\text{w} \right]. \tag{3}$$

However, do the weak values of the spin and energy observables even describe the behavior of the intensity for small interaction strengths $\alpha$? We will discuss one particular criticism on the basis of a weak DC spin rotation in the reference beam of path II, which is supposed to determine the value of $\left| \langle \hat{\sigma}_x^\text{DC} \hat{\Pi}_\text{II} \rangle_\text{w} \right|$. The according intensity is given by

$$I_\text{II}^\text{DC}(\chi_1) = \left| \langle \text{f}_\text{3path} | \text{i}_\text{3path} \rangle \right|^2 \left[ 1 + \alpha \text{Im} \left\{ \langle \hat{\sigma}_x^\text{DC} \hat{\Pi}_\text{II} \rangle_\text{w} \right\} + \frac{\alpha^2}{4} \left( \left| \langle \hat{\sigma}_x^\text{DC} \hat{\Pi}_\text{II} \rangle_\text{w} \right|^2 - \langle \hat{\Pi}_\text{II} \rangle_\text{w} \right) + \mathcal{O}(\alpha^3) \right]. \tag{4}$$

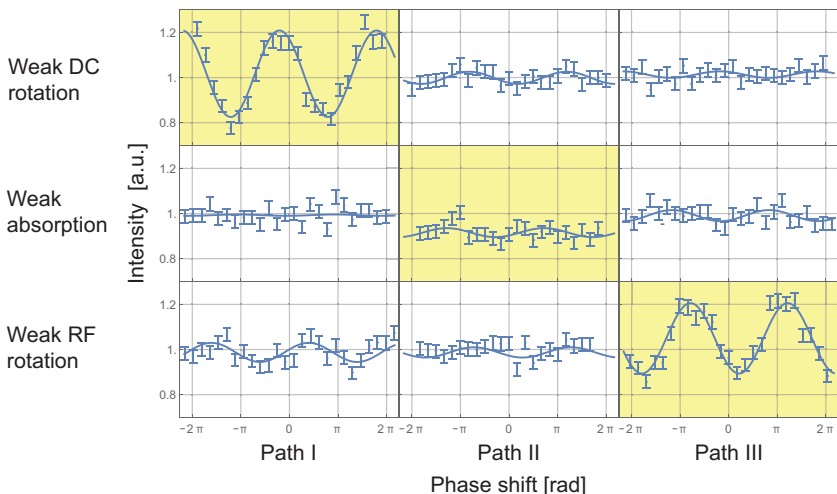

**Figure 9.** Interferograms with applied weak interactions. Intensities in O-beam given over the phase shifts induced in the path indicated at the bottom. The weak interaction is specified to the left. Conspicuous differences to the interferograms with only the preparation applied are highlighted with yellow background. As this is the case for a different path for each interaction, the perception of a three-path quantum Cheshire Cat may be given.

The first order term in $\alpha$ is connected to a sinusoidal intensity oscillation in dependence of the phase shifter orientation $\chi_1$. The intensity oscillation determines the contrast, which is linear in $\alpha$ to the lowest order. The second order terms in $\alpha$ give a constant intensity change for all $\chi_1$. In the preceding experiment [33], the authors extracted the weak values solely through the intensity at $\chi_1 = 0$ where $\text{Im}\{\langle\hat{\sigma}_x^{\text{DC}}\hat{\Pi}_{\text{II}}\rangle_w\} = 0$. Thus, the linear term in $\alpha$ vanishes and only the quadratic ones remain. Since $\langle\hat{\Pi}_{\text{II}}\rangle_w = 1$ (see [35]), the latter of the quadratic terms implies that there is some change in the mean intensity as a reaction to the unitary spin rotation in this path, as pointed out by Stuckey et al.[45]. They argue that this indicates that the properties are not separated inside the interferometer because the value $\left|\langle\hat{\sigma}_x^{\text{DC}}\hat{\Pi}_{\text{II}}\rangle_w\right|$ becomes only zero by accounting for the effect on the intensity $\propto \langle\hat{\Pi}_{\text{II}}\rangle_w$. Thereby, a measurable intensity change through a weak magnetic field is connected to a spin weak value of zero in this case. In other words, the extracted value of the spin weak value depends on the path weak value.

In answer to this dilemma, Stuckey et al. propose an additional condition for a quantum Cheshire Cat, namely that the extraction of the weak values must be solely through the interaction terms linear in $\alpha$. This is achieved with a different data analysis compared to the initial experiment of Denkmayr et al.[33]. In the present experiment as well as in above delayed-choice experiment [34], the weak value in question is not extracted from the intensity at a relative phase $\chi_1 = 0$ between the paths but from the linear effect on contrast in the cases of the spin and energy manipulations. Hence, the quadratic terms that change the mean intensity can in principle be omitted for small $\alpha$, while an extraction of the spin weak value is still possible through the linear dependence in $\alpha$ in Equation (4). This gives a similar behavior as for the path weak values in Equation (3). In this manner, the extraction of the weak values of the spin (and energy) observables is independent of the extraction of the path weak values. Therefore, the criticism by Stuckey et al. should be dispelled by means of this different method of data analysis such that the weak values describe the system for small interaction strengths.

### 2.4. Exchange of Grins in Photonic System

An example for an implementation of a quantum Cheshire Cat with photons is found in Ref. [46]. A vertically polarized 406.7 nm laser pumps a type-I cut $\beta$-barium borate (BBO) crystal. The emerging pair of photons with a wavelength of 813.4 nm is entangled in their path and spin degrees of freedom. The Franson interferometer [47] downstream of the entanglement is depicted in Figure 10. The right red beams are identified with one Cheshire

Cat, called Anna (A), and the left green beams with a second Cheshire Cat, called Belle (B). This is the authors' choice and one could just as much associate Anna with the upper and Belle with the lower beams. The upper beams are denoted by u and the lower beams by d. Each pair of upper and lower beams are recombined at the beam splitter (BS). The pre-selected state of the superposition of two entangled photon pairs is given as

$$|\xi\rangle = \frac{1}{\sqrt{2}}\left(-|\phi^-\rangle \otimes |u^A d^B\rangle + |\phi^+\rangle \otimes |d^A u^B\rangle\right) \tag{5}$$

and the post-selected state as

$$|\zeta\rangle = \frac{1}{2}\left(|\uparrow^A\rangle + |\downarrow^A\rangle\right)\left(|\uparrow^B\rangle + |\downarrow^B\rangle\right) \otimes |\psi^-\rangle \tag{6}$$

with the Bell states

$$|\phi^\pm\rangle = \frac{1}{\sqrt{2}}\left(|\uparrow^A\uparrow^B\rangle \pm |\downarrow^A\downarrow^B\rangle\right) \tag{7}$$

and

$$|\psi^\pm\rangle = \frac{1}{\sqrt{2}}\left(|u^A d^B\rangle \pm |d^A u^B\rangle\right). \tag{8}$$

Weak interactions are implemented before recombination at the beam splitter (BS) with absorbing neutral (ND) and polarizing density filters (PD). The reactions of the detected intensity are used to infer the "locations of the photons or their spins". The neutral density filters cause reactions of the detected intensity when implemented in beams $u^A$ and $d^B$. Therefore, one may conclude that the body of Cheshire Cat Anna went through the upper path while Belle went through the lower beam. The weak polarizing density filters result in spin weak values of 1 when implemented in beams $u^B$ and $d^A$. Therefore, one may conclude, that "the spins of Anna and Belle went through a different path then their bodies". The authors of Ref. [46] never state their subsequent reasoning explicitly. However, the related theoretical proposal by Das et al. [48] argues that at the beam splitter "recombination of the photons and their polarizations do occur, but in the case of the above mentioned postselected states, [Anna's] photon recombined with [Belle's] polarization while [Belle's] photon recombined with [Anna's] polarization. Thus we have exchanged the grins of two quantum Cheshire cats." A sketch of the corresponding impression is given in Figure 11. With a different choice of which beams to identify with each Cheshire Cat, the "spins recombine with the same photon" as they were associated with initially.

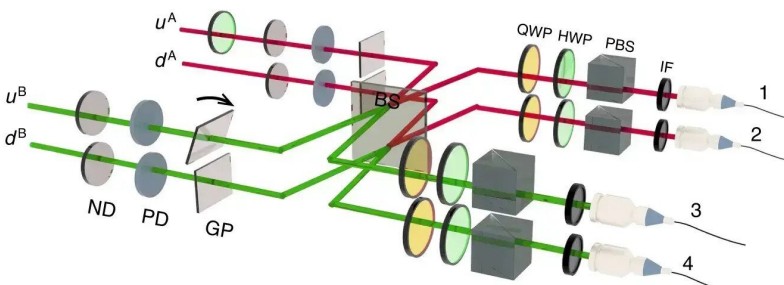

**Figure 10.** Schematic of the setup for "the exchange of grins" in a photonic system. The subfigure 2c of "Design of experiments" by Liu et al. [46], is licenced under CC BY 4.0, see http://creativecommons.org/licenses/by/4.0/ (accessed on 13 May 2023). Two entangled photons and their spins are interfered in a Franson interferometer. A and B represent the two Cheshire Cats Anna and Belle. The two heights of the paths are indicated as up (u) and down (d). The setup contains neutral density filters (ND), polarization-sensitive density filters (PD), glass plates (GP), a beam splitter (BS), quarter-wave plates (QWP), have-wave plates (HWP), polarizing beam splitters (PBS) and interference filters (IF). The intensity is recorded at four output ports.

The entanglement clearly places the observed phenomenon in the realm of quantum mechanics. In a realistically inspired interpretation of the obtained weak values and with the authors' choice for the beams of Anna and Belle, this quantum Cheshire Cat not only exhibits "the separation of particle and spin property" but potentially even the "permanent exchange of this property between two photons".

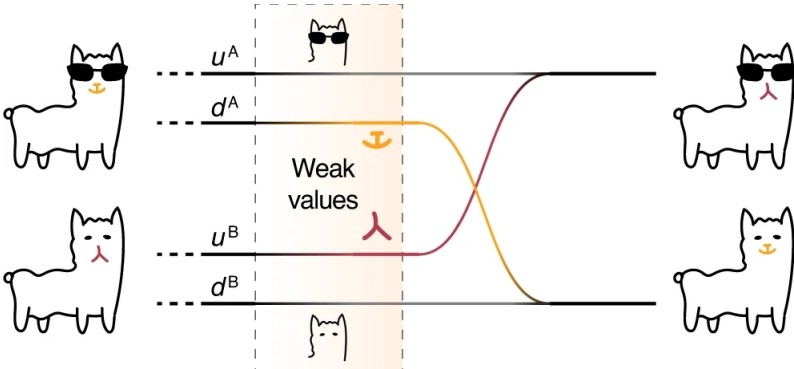

**Figure 11.** Schematic of the exchange of grins between two quantum Cheshire Cats. The figure "Schematic illustration" by Liu et al. [46] is licenced under CC BY 4.0, see http://creativecommons.org/licenses/by/4.0/ (accessed on 13 May 2023). No changes were made. Two quantum Cheshire Cats, Anna (A) and Belle (B), exchange their grin and frown. This is a possible interpretation of observations in the presented photonic experiment. Two entangled photons and their spin "appear to exchange their spins". The two heights of paths in the interferometer are indicated as up (u) and down (d).

### 3. Path Presence

Weak values can be used to investigate which path a particle takes in a double slit configuration. The particles are faintly marked in each path of an interferometer by applying a weak coupling to a pointer state, which is different in each path. The analysis of the pointer states in the output ports of the interferometer then yields information about which paths the particles have passed through. Photons for example have been marked by a reflection on a mirror that is vibrating with a distinct frequency [49], which is schematically illustrated in Figure 12a. In response to this experiment, neutrons have been marked by a small energy change induced by radio-frequency (RF) coils using different frequencies in each path of a three-path interferometer [50] (see Figure 12b), or—even simpler—by applying slightly different spin rotations in each path of a two-path interferometer [51]. The presence of the particles in a particular path is then given by the real part of the path projection operator's weak value, obtained by a weak measurement. The coupling to the pointer state is so weak that the disturbance on the system is negligible. In particular, the visibility of the interference fringes is maintained. Since the visibility is complementary to the obtainable which-path information [52], only little information can be gained per event. However, by collecting the information over many measurements, the weak value can be eventually determined.

Being measured on a large ensemble, the weak value represents an average over the ensemble. Can we tell anything about the presence of an individual particle? The weak measurement gives the average value as well as the variance of this value, and this variance is usually large. Imagine a spin prepared in $+x$ direction that is slightly rotated by an angle $\alpha$ towards $+y$ in one path of the interferometer for path marking. To be the most sensitive to the $y$ component, we analyze the spin in the $\pm y$ direction in the interferometer output ports. In fact, the path presence $\omega$, i.e., the real part of the weak value of the path projection operator, is in the simplest case given by the spin expectation value in $y$ direction divided by the rotation angle, $\omega \approx \langle \sigma_y \rangle / \alpha$. However, since the spin direction is still close to the initial $+x$ direction, the analysis in the $y$ direction will give nearly equal probability for the $+y$ and

$-y$ outcomes, and the variance of $\langle\sigma_y\rangle$ will be close to its maximum. Therefore, the obtained presence of the particles is only an average and cannot be attributed to individual particles.

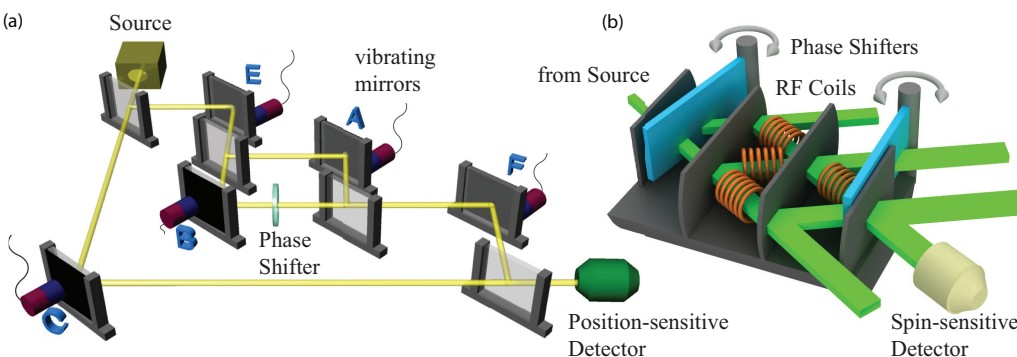

**Figure 12.** Schematic sketch for which-way measurements using faint traces induced by (**a**) vibrating mirrors for photons from 2013 [49] and (**b**) small energy kicks via RF coils for neutrons from 2018 [50]. Gray arrows indicate the spatial rotation of the two phase shifters.

To circumvent this problem, Hofmann [53] proposed the method of feedback compensation. Instead of analyzing the spin in the $y$ direction, we apply a back rotation by an angle $\beta$ (the "compensation"), which restores the initial spin state. Importantly, under certain conditions, there is an optimal compensation angle $\beta_0$, which allows us to restore all spins to the $+x$ state. Therefore, when analyzing the spin in the $x$ direction, the variance vanishes and this means that the average value is valid for each individual neutron. The ratio $\beta_0/\alpha$ of the rotation angles directly gives the real part of the weak value. If, for example, a compensation angle of $\beta_0 = 2/3\,\alpha$ is necessary to reverse the original $\alpha$ rotation applied in path I, we can conclude that the neutron had a presence of $\omega = 2/3$ in path I. Being a weak value, the path presence depends on the final state, which is given in our case by one or the other exit beam of the interferometer. For optimal compensation, information about the final state is required (the "feedback") which means that we have to apply different compensation angles in the two exit beams, cf. Figure 13.

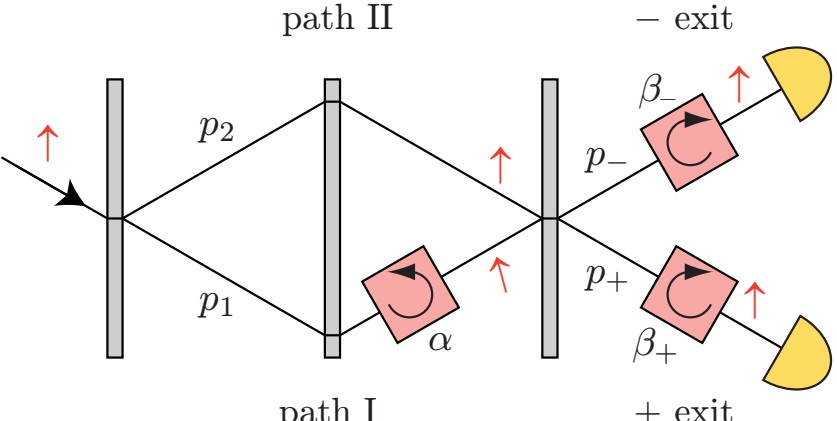

**Figure 13.** Feedback compensation scheme applied to a path measurement in a neutron interferometer. The black arrow on the left is the incident beam direction. The circular black arrows give the orientation of spin rotations in the quadratic spin manipulators. Red arrows indicate the local polarization vectors during the procedure. Neutrons are initially polarized in $+x$ direction, denoted by the up arrow. The spin is rotated in path I by a small angle $\alpha$. The "compensation" rotations by $\beta_\pm$ in the exit beams can fully restore the original spin orientation.

The experiment [36] was performed at the neutron interferometry station S18 at the ILL, Grenoble. The optimal compensation values $\beta_{0\pm}$ were determined by trial and error. This corresponds to a conventional weak measurement and provides the weak value as

an average value. However, once this value is known, it can be verified on subsequent neutrons. Again, an ensemble of neutrons is needed to test the effect of a particular compensation value. If it turns out that all spins end up in the $+x$ state, the variance vanishes, meaning that the average value equals the individual values. One can therefore conclude that the obtained path presence was indeed valid for each individual neutron. The theoretical framework of verifying an estimate has been elaborated by Hall [54], described further below in more detail. Based on Ozawa's universal uncertainty relation [55], he showed that the verification of an estimate can in principle be error-free, even without disturbing the system.

In the experiment, the focus was on the simplest case where the interferometer phase is adjusted to zero. Then, the forward exit labeled $|+\rangle$ represents constructive interference, $|+\rangle = |\mathrm{I}\rangle + |\mathrm{II}\rangle$, and the side exit labeled $|-\rangle$ represents destructive interference, $|-\rangle = |\mathrm{I}\rangle - |\mathrm{II}\rangle$, and all weak values are real values. In addition, the first beam splitter of the interferometer was prepared with an asymmetry of 4:1 to avoid vanishing intensities due to totally destructive interference. The calculated path presences of the prepared state $|\psi\rangle = a_1|\mathrm{I}\rangle + a_2|\mathrm{II}\rangle$ are summarized in Table 1. Neutrons reach the $|+\rangle$ exit with probability $p_+ = 90\%$ and the $|-\rangle$ exit with probability $p_- = 10\%$. Neutrons that end up in the $|+\rangle$ exit had a path presence of $\omega_{1+} = 2/3$ in path I and of $\omega_{2+} = 1/3$ in path II, which corresponds to the amplitude ratio of the preparation. However, neutrons ending up in the $|-\rangle$ exit had a presence of $\omega_{1-} = 2$ in path I and $\omega_{2-} = -1$ in path II. These are anomalous weak values lying outside of the eigenvalue spectrum. Note that they still sum up to unity, $\omega_{1-} + \omega_{2-} = 1$. In addition, the initial probabilities $p_j$ in either path $j$ are reproduced by the weighted average $\bar{\omega}_j$ of the weak values. This is evident, since the probabilities $p_j$ are actually the expectation values $\langle \hat{\Pi}_j \rangle$ of the path projection operators and can be equally expressed by summing over the eigenstates $|\mathrm{I}\rangle$ and $|\mathrm{II}\rangle$ using the eigenvalues $\lambda_1 = 1$ and $\lambda_2 = 0$ or by summing over the final states $|+\rangle$ and $|-\rangle$, which also represent a complete basis.

**Table 1.** Path presences for a 4:1 beam splitter. (**a**) Preparation, and (**b**) path presences depending on the final state.

| (a) | | Path I | Path II |
|---|---|---|---|
| initial amplitudes | | $a_1 = \frac{2}{\sqrt{5}}$ | $a_2 = \frac{1}{\sqrt{5}}$ |
| initial probabilities | | $p_1 = \frac{4}{5}$ | $p_2 = \frac{1}{5}$ |

| (b) | Probability | Presence in Path I | Presence in Path II |
|---|---|---|---|
| + exit | $p_+ = \frac{9}{10}$ | $\omega_{1+} = \frac{2}{3}$ | $\omega_{2+} = \frac{1}{3}$ |
| − exit | $p_- = \frac{1}{10}$ | $\omega_{1-} = 2$ | $\omega_{2-} = -1$ |
| average | | $\bar{\omega}_1 = \frac{4}{5}$ | $\bar{\omega}_2 = \frac{1}{5}$ |

$$\langle \psi | \hat{\Pi}_1 | \psi \rangle = |\langle \psi | \mathrm{I} \rangle|^2 \, \lambda_1 \; + |\langle \psi | \mathrm{II} \rangle|^2 \, \lambda_2 \quad = p_1 \, \lambda_1 \; + p_2 \, \lambda_2 \quad = p_1 \tag{9}$$

$$= |\langle \psi | + \rangle|^2 \, \omega_{1+} + |\langle \psi | - \rangle|^2 \, \omega_{1-} = p_+ \, \omega_{1+} + p_- \, \omega_{1-} = \bar{\omega}_1 \tag{10}$$

One can see that the weak values $\omega_{1\pm}$ play the role of the eigenvalues $\lambda_{1,2}$ in case the state is not measured in the operator's eigenbasis but in another (complete) set of states. The preparation alone, i.e., the asymmetric beam splitter, determines the presence of the neutrons in path I only in a statistical sense by the probability $p_1$. Only if the final state is known, a more detailed statement about individual neutrons can be made. If we tried to detect the neutrons directly in path I, we would divide the neutrons into the two sub-ensembles "path I" and "path II" and would obtain a path-I presence of $\lambda_1 = 1$ for the former and $\lambda_2 = 0$ for the latter sub-ensemble. The experiment divides the neutrons into the sub-ensembles "+exit" and "−exit" and the obtained path-I presences are $\omega_{1+}$ and

$\omega_{1-}$, respectively. In any case, the path presence can only be attributed in retrospect once the neutron has been detected.

The complete experimental setup is depicted in Figure 14. All path presences listed in Table 1 were confirmed. While large rotation angles $\alpha$ allowed for precise measurements of the ratio $\beta_0 / \alpha$, the theoretical weak values were only approached in the limit of small $\alpha$, cf. Figure 15. In a previous experiment [56], the path presence was measured by applying an $\alpha/2$ rotation in both paths but in opposite directions, which is an equivalent way to mark the path. The novelty of the current experiment is the extension of the weak measurement to the feedback compensation scheme.

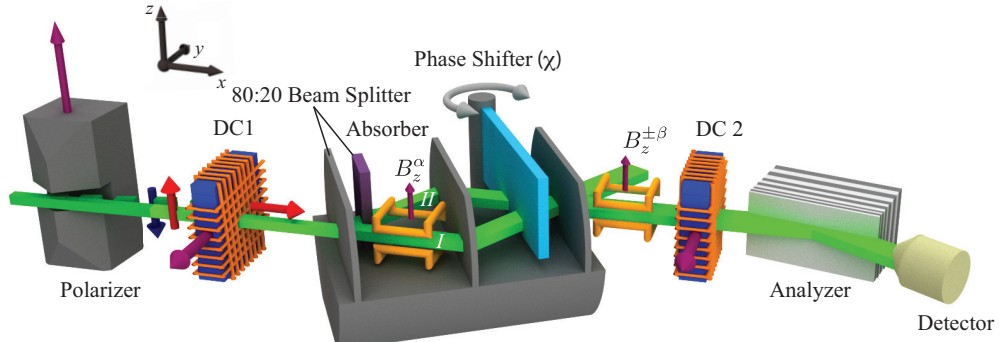

**Figure 14.** Experimental setup of the feedback compensation scheme shown in Figure 13. Purple arrows give the direction of local magnetic fields, and gray arrows indicate the spatial rotation of the phase shifter. Red and blue arrows are the initial up and down spin polarization vectors. The fraction that is initially up polarized is rotated by the field in the coil DC1 into the $x$ direction. The spin analysis is implemented only in one of the exit beams. This exit represents the $|+\rangle$ exit if $\chi = 0$ and the $|-\rangle$ exit if $\chi = \pi$.

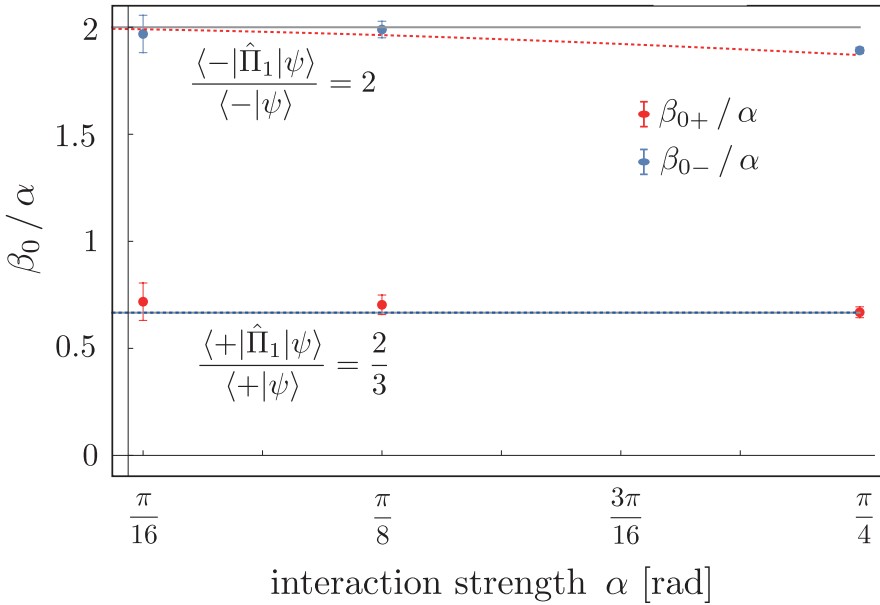

**Figure 15.** Experimental results of the path presence in path I, given by $\omega_{1\pm} \equiv \lim_{\alpha \to 0} \beta_{0\pm} / \alpha$, versus interaction strength $\alpha$ for $\alpha = \pi/4$, $\pi/8$ and $\alpha = \pi/16$.

Furthermore, the experiment allows for an efficient determination of the so-called Ozawa–Hall error $\varepsilon$, which describes the accuracy of a quantum measurement. The Ozawa–Hall error $\varepsilon$ is a central element in the universally valid reformulation of Heisenberg's uncertainty principle, a so-called measurement uncertainty relation. Unlike preparation uncertainty relations, which set limits on how sharp the values of two observables can be

determined if measured separately, measurement uncertainty relations provide information of the error when measuring one observable and the thereby induced disturbance on another subsequently (or simultaneously) measured observable—in other words, they characterize the joint measurement. The universally valid uncertainty relation is a certain type of these measurement uncertainty relations, which reads

$$\varepsilon_A \eta_B + \varepsilon_A \Delta_B + \eta_B \Delta_A = \frac{1}{2} |\langle\psi|[A, B]|\psi\rangle|, \tag{11}$$

where $\varepsilon_A$, $\eta_B$, $\Delta_A$ and $\Delta_B$ denote the error of the first measurement, the thereby induced disturbance on the second measurement and the variances of $A$ and $B$ [55]. In our case, the observable $A = \hat{\Pi}_1$ is represented by the path presence inside the interferometer and $B = \hat{\sigma}_x$ depends on whether the interfering components leave the interferometer in a forward or reflected direction. Due to the particular experimental configuration, the measurement of the path presence does not disturb the interference. This is reflected in a vanishing disturbance $\eta_B = 0$, resulting in a reduced form of the universally valid uncertainty relation denoted as $\varepsilon_A \Delta_B = 1/2|\langle\psi|[A, B]|\psi\rangle|$, with standard deviation $\Delta_B = \Delta(\hat{\sigma}_x, |\psi\rangle) = 3/5$.

The error $\varepsilon_A$ is given by the statistical deviation between the operator of interest, namely $\hat{\Pi}_1$, and the estimated value of that operator, which corresponds to a feedback of $\beta_{\pm}$ that can compensate a weak interaction of $\alpha$, given by $\beta_{\pm}/\alpha$. In our case, it reads

$$\begin{aligned} \varepsilon^2(\hat{\Pi}_1) \quad &= \sum_{\pm} \langle\psi| \left(\hat{\Pi}_1^\dagger - \frac{\beta_{\pm}}{\alpha}\right)|\pm\rangle\langle\pm| \left(\hat{\Pi}_1 - \frac{\beta_{\pm}}{\alpha}\right)|\psi\rangle \\ &= \sum_{\pm} p_{\pm} \left|w_{1\pm} - \frac{\beta_{\pm}}{\alpha}\right|^2 \end{aligned} \tag{12}$$

where $p_{\pm} = |\langle\pm|\psi\rangle|^2 = \frac{1}{2} \pm a_1 a_2$ denotes the statistical probability of finding the neutron in the final state $|\pm\rangle$, with initial amplitudes $a_1$ and $a_2$ of the beams in each path. This means, the uncertainty vanishes completely if the compensations $\beta_{\pm}/\alpha$ applied in the output ports $|+\rangle$ and $|-\rangle$, respectively, equal the corresponding weak values $\omega_{1\pm}$. Then, the compensations are no longer just estimates but precise measurements of the weak values with an experimentally determined error of $\varepsilon^2(\hat{\Pi}_1) = 0.0012(29)$, which is in good agreement with the theoretical predicted value of zero from Equation (12).

To conclude, the experiment shows that every individual neutron is distributed over both paths in an exactly quantifiable ratio. This result questions interpretations of quantum mechanics such as the de Broglie–Bohm interpretation, which claims that every particle goes one or the other way and that particles are only statistically distributed over both paths.

## 4. Direct Test of Commutation Relation

The canonical commutation relation is a corner stone of quantum theory. It was originally formulated by Heisenberg in 1925 [57] as $\hat{p}\hat{q} - \hat{q}\hat{p} = i\hbar \mathbb{1}$, where $\hat{p}$ and $\hat{q}$ are the matrix forms of position and momentum variables. Only two years later, Heisenberg proposed his famous uncertainty relation $\delta p \delta q \sim h$, which he regarded as a direct mathematical consequence of the canonical commutation relation. Here, $\delta p$ and $\delta q$ represent uncertainties in momentum and position measurements, respectively. Later, building upon Kennard's [58] idea of interpreting the uncertainties as standard deviations, Robertson [59] generalized Heisenberg's preparation uncertainty relation for any two arbitrary observables $A$ and $B$ so that $\Delta A \Delta B \geq 1/2|\langle\psi|[A, B]|\psi\rangle|$. In recent times, the distinction between preparation and measurement uncertainty relations has been made and many interesting new formulations have also been proposed [60–66], including Ozawa's universally valid uncertainty relation, given by Equation (11).

The first experimental test of the non-commutativity of Pauli spin matrices utilized different sequences of rotations (on the same initial state), which is schematically illustrated for neutrons and photons in Figure 16. The neutron interferometric experiment reported in [67] was performed at the Reactor of the University of Missouri (MURR), USA, by the former BARC-Vienna-MURR collaboration.

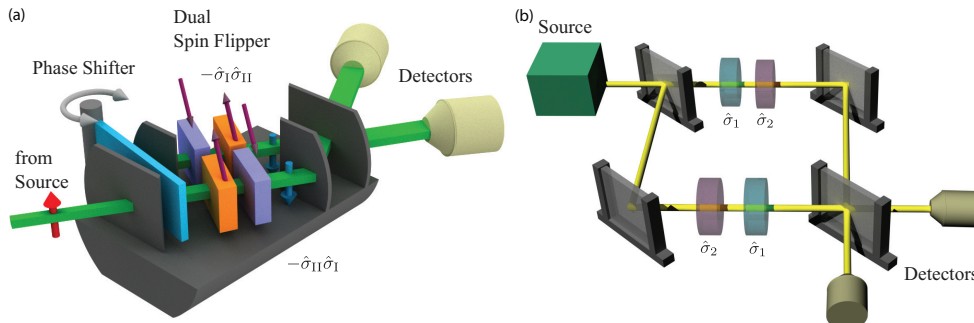

**Figure 16.** Schematic sketch for the experimental test of the non-commutativity of Pauli spin matrices using different sequences of rotations (**a**) for neutrons from 1997 [67] and (**b**) for photons from 2010 [68]. The gray arrow indicates the spatial rotation of the phase shifter. The red arrow indicates the neutron's initial up spin state. Purple arrows indicate the orientation of local magnetic fields.

However, a direct test of the commutation relation is an experimentally challenging task for the following reason: while in quantum mechanics observables are usually represented by Hermitian operators, the product of two non-commuting observables (as occurring in the commutator relation) is in general non-Hermitian. Consequently, a direct test of the canonical commutation relation should unambiguously determine the expectation value of the non-Hermitian product of two non-commuting observables as it occurs in the commutation relation. The experimental realization for a direct test of the commutation relation is therefore conceptionally different compared to prior experiments. The experiment was carried out at the neutron interferometry station S18 at the ILL, Grenoble, and is reported in [37]. In the neutron interferometric experiment, a measurement of a single anomalous weak value of a relevant path-qubit observable is performed, where the coupled spin-energy degree of freedom serves as ancilla.

Non-Hermitian observables can be expressed in terms of weak values as seen from

$$\langle\psi|[\hat{A},\hat{B}]|\psi\rangle \;=\; 4\langle\psi|\hat{\Pi}_A^+\hat{\Pi}_B^+|\psi\rangle - 4\langle\psi|\hat{\Pi}_B^+\hat{\Pi}_A^+|\psi\rangle = -8\mathrm{i}|\langle+_B|\psi\rangle|^2\mathrm{Im}\left\{\langle\hat{\Pi}_A^+\rangle_{\mathrm{w}}^{\psi,+_B}\right\} \tag{13}$$

where $\langle\hat{\Pi}_A^+\rangle_{\mathrm{w}}^{\psi,+_B}$ is the weak value of the projector $\hat{\Pi}_A^+$ for a post-selected state $|+_B\rangle$ and pre-selected state $|\psi\rangle$, while $|\langle+_B|\psi\rangle|^2$ is the probability of successful post-selection. For Pauli spin matrices $\hat{A}=\hat{\sigma}_z$ and $\hat{B}=\hat{\sigma}_x$, the commutation relation is given in the form of

$$\langle\psi|(\hat{\sigma}_z\hat{\sigma}_x - \hat{\sigma}_x\hat{\sigma}_z)|\psi\rangle = 2\mathrm{i}\langle\psi|\hat{\sigma}_y|\psi\rangle. \tag{14}$$

Using $\hat{\sigma}_x = 2\hat{\Pi}_x^+ - \mathbb{1}$, $\hat{\sigma}_y = 2\hat{\Pi}_y^+ - \mathbb{1}$ and $\hat{\sigma}_z = 2\hat{\Pi}_z^+ - \mathbb{1}$, where we have $\hat{\Pi}_x^+ = |+_x\rangle\langle+_x|$, $\hat{\Pi}_y^+ = |+_y\rangle\langle+_y|$ and $\hat{\Pi}_z^+ = |+_z\rangle\langle+_z|$, the Pauli spin matrices $\hat{\sigma}_x$, $\hat{\sigma}_y$ and $\hat{\sigma}_z$ are expressed in terms of projectors. Combining this with Equation (13) and using the definition of the weak value, we obtain

$$4|\langle+_x|\psi\rangle|^2\,\mathrm{Im}\left\{\langle\hat{\Pi}_z^+\rangle_{\mathrm{w}}^{\psi,+_x}\right\} = -2|\langle+_y|\psi\rangle|^2 + 1, \tag{15}$$

where $\langle\hat{\Pi}_z^+\rangle_{\mathrm{w}}^{\psi,+_x} = \langle+_x|\hat{\Pi}_z^+|\psi\rangle/\langle+_x|\psi\rangle$ is the weak value of $\hat{\Pi}_z^+$, that is the projection operator onto the state $|+z\rangle$, given the pre- and post-selected states $|\psi\rangle$ and $|+_x\rangle$, respectively. Thus, the imaginary part of a single weak value $\langle\hat{\Pi}_z^+\rangle_{\mathrm{w}}^{\psi,+_x}$ allows to evaluate the left-hand side (LHS) of the commutation relation given by Equation (14). For determining the quantity on the right-hand side (RHS), it is necessary to separately measure the post-selected probability $|\langle+_y|\psi\rangle|^2$.

To experimentally test Equation (15), the setup depicted in Figure 17 is applied. The system is pre-selected in the *path* state $|\psi_{\mathrm{i}}(\chi)\rangle = \frac{1}{\sqrt{2}}(|\mathrm{I}\rangle + e^{-\mathrm{i}\chi}|\mathrm{II}\rangle)$ and post-selected in the state $|\psi_{\mathrm{f}}\rangle \equiv |+_x\rangle$, with $|+_x\rangle = (|\mathrm{I}\rangle + |\mathrm{II}\rangle)/\sqrt{2}$, where $|\mathrm{I}\rangle$ and $|\mathrm{II}\rangle$ denote the eigenstates of the Pauli (path) observable given by $\hat{\sigma}_z = |\mathrm{I}\rangle\langle\mathrm{I}| - |\mathrm{II}\rangle\langle\mathrm{II}|$.

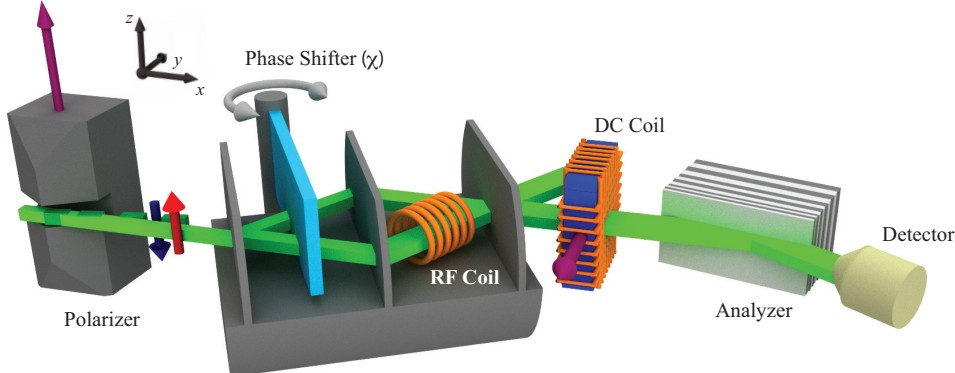

**Figure 17.** Setup of the experiment for the commutation relation. The purple arrows give the directions of local magnetic fields and the gray arrow indicates the spatial rotation of the phase shifter. Red and blue arrows are the initial up and down spin polarization vectors. In path I, the neutron beam passes through a radio-frequency (RF) spin flipper, where the combined spin/energy degree of freedom is used as a marker.

In the actual weak measurement of $\hat{\Pi}_1$, the weak interaction is implemented by use of the RF spin manipulator in path I inducing a (small) spin-rotation of an angle $\alpha = \pi/9$ at a frequency of $\omega/2\pi = 60$ kHz ($\alpha$ is directly dependent on the magnetic field strength of the oscillating magnetic field). The sufficiently small value of $\alpha$ warrants the required weak measurement criteria. As explained in detail in [37], the weak value $\langle \Pi_1 \rangle_w^{\psi_i,+x}$ is extracted from the time-dependent intensity oscillation observed at the detector that is given by

$$I(t,\chi) = \frac{1}{2}|\langle +_x|\psi_i(\chi)\rangle|^2 \left(1 + \alpha \mathrm{Im}\left\{e^{i\omega t}\langle \hat{\Pi}_1 \rangle_w^{\psi_i,+x}\right\}\right). \tag{16}$$

The resulting combination of all three measurements, namely $|\langle +_x|\psi_i(\chi)\rangle|^2$, $|\langle +_y|\psi_i(\chi)\rangle|^2$ and $\mathrm{Im}\left\{\langle \Pi_1 \rangle_w^{\psi_i,+x}\right\}$, is given in Figure 18. The measurement results accounting for the LHS and RHS of the commutation relation in Equation (15) are represented by orange and green data points, respectively. The result is in good agreement with the relevant theoretical prediction (dotted blue), experimentally verifying the canonical commutation relation for Pauli spin matrices, as expressed in Equation (14).

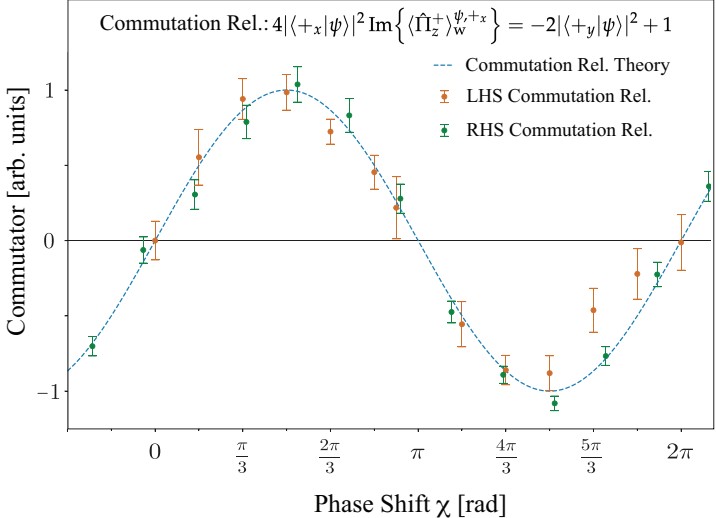

**Figure 18.** Experimental results of left-hand side (orange) and right-hand side (green) of the commutation relation in Equation (15) are plotted as a function of the phase shift $\chi$.

## 5. Discussion

From the mathematical side, it is evident that weak values depend on both the pre-selection of the initial quantum state and the post-selection of a certain final state. What this means for our understanding of physics is still the subject of discussions. It is suggested that weak values provide elements of reality between pre- and post-selection, which are *inferred* with certainty as measurement results of a physical quantity and can have unusual and counterintuitive properties [69]. This might be surprising since the value of an observable is in principle undetermined until a projective measurement is performed. However, there is no contradiction because weak values depend on the post-selection, which represents the actual measurement here. Aharonov et al. proposed a time-symmetric interpretation of quantum mechanics, called two-state vector formalism, where the system receives information from both the past (the pre-selection) and the future (the post-selection) [70]. Our experiments cannot decide on this question, but they clearly support the following more conventional point of view. The post-selection divides the initial ensemble into sub-ensembles. In some experiments, only one particular sub-ensemble is investigated, but in principle one can always regard all of them to obtain the complete picture. Each sub-ensemble has its own weak values. As already argued in the path presence experiment (Section 3), averaging the weak values over all sub-ensembles gives again the expectation value of the observable, which depends on the initial state alone. In the Cheshire Cat experiment (Section 2.1), the neutron beam is initially evenly distributed over both interferometer paths. If we regard a certain post-selected ensemble, we notice that the neutrons of this sub-ensemble were sensitive to absorption only in one path and sensitive to spin manipulation only in the other path. However, if we regard another post-selected ensemble, the situation might be completely different, e.g., the roles of the interferometer paths might be swapped. This is exactly what we observe in the delayed-choice Cheshire Cat experiment (Section 2.2). The weak values describe the reactions of each individual sub-ensemble to weak interactions. At the time of interaction, the particles are still in a state of superposition of belonging to one or the other sub-ensemble. Once the post-selection is performed, the sub-ensemble is determined and one can in retrospect tell for each particle what happened between pre- and post-selection. The method of feedback compensation even shows that weak values represent not only averages over all particles of a sub-ensemble but can in certain cases indeed be attributed to individual particles.

## 6. Conclusions

We have shown recent achievements of neutron interferometer experiments, which investigate variants of the quantum Cheshire Cat phenomenon, the measurement of the neutron's presence in the paths of the interferometer as well as a commutation relation. Note that all experiments take advantage of weak measurements; they make reactions to minimal disturbances of the quantum states under investigation accessible, which could not be achieved with conventional measurements. Using weak interactions allows us to look into details of quantum dynamics. The theoretical framework of the weak measurement and the weak value, a kind of quantum variable representing the intermediate response of a quantum system, was introduced by Aharonov and his co-workers [27], followed by successive experimental performances [51]. In the quantum Cheshire Cat experiments, these responses are used to "assign the locations of the neutrons and their properties". The first experiment concerning the effect of the quantum Cheshire Cat clearly demonstrates that the system behaves as if the neutrons go through one beam path, while their magnetic moment "travels along the other path" [33]. The following delayed-choice experiment allows us to resolve the fact that the behavior of the neutron and its spin are fully determined by the choice of the selection at a later time; quantum-mechanical causality is implemented [34]. The last experiment studying the effect of the quantum Cheshire Cat with the three-path neutron interferometer gives a more detailed proposition of the origin of the apparent behavior of the quantum system [35]; distinct responses to the weak interactions are due to the filtration procedure of the post-selection applied on the

states with and without weak interaction. Furthermore, it is confirmed in the path presence experiment that a fraction of the neutron is present in the paths of the interferometer; the obtained path presence is attributed to (the real part of) the weak value of the path projector and is not given by a statistical average but applies to every individual neutron. This shows that every individual neutron is distributed over both paths in an exactly quantifiable ratio. The obtained result questions interpretations of quantum mechanics such as the de Broglie–Bohm interpretation, which claims that every particle goes one or the other way and that particles are only statistically distributed over both paths. The experiment concerning the commutation relation gives direct experimental evidence for a relation primarily used in calculus.

**Author Contributions:** A.D. and Y.H. coordinated the collaboration; all authors co-wrote the paper. All authors have read and agreed to the published version of the manuscript.

**Funding:** This research was supported by the Austrian science fund (FWF) Projects No. P 30677, P 34105 and P 34239. Y.H. is partly supported by KAKENHI Project No. 18H03466.

**Data Availability Statement:** No new data were created or analyzed in this study. Data sharing is not applicable to this article.

**Acknowledgments:** We would like to express our gratitude for the kind hospitality at the ILL in Grenoble and the ongoing support of the science division at ILL.

**Conflicts of Interest:** The authors declare no conflict of interest.

## Abbreviations

The following abbreviations are used in this manuscript:

| | |
|---|---|
| Abs | absorber |
| BARC | Bhabha Atomic Research Centre |
| BBO | barium borate |
| DC | direct-current |
| ILL | Institut Laue-Langevin |
| LHS | left-hand side |
| MURR | University of Missouri Research Reactor Center |
| PS | phase shifter |
| RF | radio-frequency |
| RHS | right-hand side |
| SR | spin-rotator |
| USA | United States of America |

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
