# Peer review of "Neutron Interferometer Experiments Studying Fundamental Features of Quantum Mechanics"

_atoms, doi:10.3390/atoms11060098_

Round 1

Reviewer 1 Report

My comments and suggestions are in the provided pdf file 'Weak Measurements at the ILL'

Reviewer 2 Report

The paper is a good one, reporting and summarizing an interesting research. I have, however, a series of several (mostly minor) comments and suggestions, to the goal of making it a more readable and complete paper.

1) The paper, as usually happens when discussing purely quantum mechanical effects, has a philosophical (more correctly: epistemological) overtone. Some of my comments will of course be in that direction. Let me start with line 5, where I suggest to put "en route" in italics. This expression in fact refers to our common sense classical concept.

2) Line 8. I propose that you say that the "incident neutron beam is (IN THE CLASSICAL SENSE) split in two or three beams". Same reason as 1.

3) I like your conclusion at line 29 and I propose to you the addition of a reference, consisting of a well known book: Peter R. Holland - The quantum theory of motion - Cambridge University Press 1993.

4) I appreciated your list of discovery of "demonstration of double-slit experiment" in lines 31 to 37. However, I propose here two changes:

4a) The "electron" experiment was done for the first time by Merli, Missiroli and Pozzi (about 10 years before Tonomura), so I believe it is important to cite P.G. Merli, G.F. Missiroli, G. Pozzi, "On the statistical aspect of electron interference phenomena" Am. J. Phys. 44 (1976) 306.

4b) Please add to your list also the first demonstration of Antiparticle quantum interference, done with the positron (in single particle mode): S. Sala, A. Ariga, A. Ereditato, R. Ferragut, M. Giammarchi, M. Leone, C. Pistillo and P. Scampoli, Sc. Adv. 5 (2019) eaav760.

5) Your introduction is rather complete and the rest of your article is heavily based on the concept of weak measurement. In this respect, you quote the Aharonov, Albert, Vaidman 1988 paper, which is good. I believe you should also quote the more complete (and a bit philosophical) paper T.A. Brun, "A simple model of quantum trajectories", Am. J. Phys. 70 (2002) 719.

6) In the description of your Cheshire Cat experiments I feel that a minimal set of experimental information are missing: a) the neutron flux, b) the neutron velocity or energy. Is the source incoherent? Can we confidently say that (in classical terms) we are in the "one-particle-at-a-time" regime or not? Please specify, it is nice to know!

7) Lines 125 to 128, I suggest to adjust a bit the language and when using expressions like "the neutrons go through path II" please put them in quotes because you are using classical common sense intuition there (as you know).

8) Line 131 same as above

9) Line 158 same as above: the "locations" of the cat and the grin. This type of consideration applies to other parts of the paper, so I would suggest to use the quotes everywhere you mean that a classical reasoning would say that some physical quantity is located here or there.

10) Line 171 a spacebar is missing between Hilbert and space...

11) Most of the analysis is very technical and interesting and again whenever appropriated please indicate particle fluxes, energies and the like.

12) I was intrigued by your extension of the weak measurement to the feedback compensation scheme in the Path presence work. In that respect (after figure 13) I have a question. You determine the Ozawa error and at the end of your analysis you indicate that your result questions the de Broglie-Bohm interpretation. My question here is: what about the Fujikawa error? Did you consider calculating this type of error? (I am not suggesting to do it, just asking...)

13) Somebody needs to pay attention to the formula and the margin at line 387

14) I would suggest that you put your conclusion about the de Broglie-Bohm interpretation (in the Ozawa error) also in the conclusions of the paper. To me this is a relevant part of your work.

I congratulate the authors for this nice paper and of course suggest publication with minor modifications.

Rather good, only minor editing required.

Reviewer 3 Report

This paper reviews several quantum optical experiments using matter-wave neutron interferometers that, in the words of the authors ``throw light on striking and counter-intuitive aspects of quantum theory''.  Although these aspects of quantum mechanics are, or should be, well known by now, the review is well written is a significant contribution towards a deeper general understanding of quantum phenomena.

My only quibble is that the physical significance of weak measurements are often not fully understood in the literature and hence their significance sometimes overstated. This review would be a good opportunity to address this in the context of specific experiments.

Weak measurements do not actually constitute measurements of properties of a quantum system at any intermediate time between pre- and post- selection. Instead, using ensemble language, the pre-selection creates an particular quantum ensemble of particles in a given state and then the post-selection chooses only elements of the ensemble that have the post-selected attributes. This post-selection allows one to use classical intuition to infer certain properties of these elements in intermediate times. The surprise as usual is that the classical intuition is misguided. In fact, the values obtained at post-selection do not exist prior to the measurement that constitutes the post-selection. So statements such as ``the particle and spin properties can appear (my italics) to be separated in different paths...'' would be misleading with the very important word: appear. The key point as that these particular properties do not have definite values at any time during the particles passage through the interferometer, and the mistake made by applying classical intuition is to consider both properties to ``exist'' simultaneously, but in different branches. 

While most experts (but perhaps not all) know this, I would suggest that in a review such as this it might be appropriate to provide a more extensive discussion of not only the ``counter-intuitive'' nature of the results, but also an explanation of where the classical intuition goes wrong. 

I suggest that the authors consider the above remarks prior to publication of the manuscript.

Round 2

Reviewer 1 Report

I think the paper has been generally improved.  I especially appreciate the much better discussion and conclusion sections.

There are some minor edits:

Line 23 “Laser” doesn’t need capitalization

Line 37 Paragraph needs indentation

Line 44 “…passes through the …”

Line 62 Strike  “Lots of”

Line 283 Change NP to ND!
